# Adult-Onset Leukoencephalopathy with Axonal Spheroid and Pigmented Glia: Different Histological Spectrums Presented in Autopsy Cases of Siblings and a Surgical Case of Stereotactic Biopsy

**DOI:** 10.3390/diagnostics13061018

**Published:** 2023-03-07

**Authors:** Hee Jung Kwon, Duk L. Na, Hee Jin Kim, Yeon-Lim Suh

**Affiliations:** 1Department of Pathology, Yeungnam University Medical Center, Yeungnam University College of Medicine, Daegu 42415, Republic of Korea; 2Department of Neurology, Samsung Medical Center, Sungkyunkwan University School of Medicine, Seoul 06351, Republic of Korea; 3Department of Pathology and Translational Genomics, Samsung Medical Center, Sungkyunkwan University School of Medicine, Seoul 06351, Republic of Korea

**Keywords:** adult-onset leukoencephalopathy with axonal spheroid and pigmented glia, presenile dementia, *CSF1R*, autopsy, hereditary diffuse leukoencephalopathy with neuroaxonal spheroids, pigmented orthochromatic leukodystrophy

## Abstract

Adult-onset leukoencephalopathy with axonal spheroids and pigmented glia encompasses hereditary diffuse leukoencephalopathy with axonal spheroids and pigmented orthochromatic leukodystrophy. We describe the clinicopathological and genetic findings of three patients with this disorder. All patients presented with dysarthria, with or without cognitive decline. The first and second patients were siblings who died of the disease at ages 42 and 54, respectively, while the third patient has been bedridden. Brain magnetic resonance imaging revealed T2 hyperintensities in the subcortical and periventricular white matter. Pathological diagnosis was established by brain autopsy in cases 1 and 2, and a stereotactic brain biopsy in case 3, followed by genetic analysis of colony stimulating factor-1 receptor gene. A heterozygous c.2345G > A (p.R782H) variant was identified in the autopsy-proven cases, and a c.1765G > A (p.G589R) variant in the biopsy-proven case. Postmortem examination revealed severe white matter degeneration involving the bilateral frontoparietal lobes, but sparing the subcortical U-fibers. All cases revealed widespread loss of myelinated axons in the white matter lesions; however, axonal spheroids and pigmented macrophages were abundant in cases 1 and 3 and much less in case 2. Adult-onset leukoencephalopathy with axonal spheroids and pigmented glia should be considered in patients with presenile dementia and diffuse white matter lesions.

## 1. Introduction

Leukodystrophies are genetically determined disorders that affect the white matter of the central nervous system, regardless of the structural white matter component, molecular process, patient age, and disease course. There are different adult-onset leukodystrophies, and most have similar or indistinguishable clinical presentations, such as cognitive decline, personality problems, parkinsonism, and seizures, which making accurate diagnosis difficult. Genetic alterations, which are very helpful in making an accurate diagnosis of leukodystrophies, have been discovered in genetic studies. The classification of adult-onset leukoencephalopathies based on causative genes was compiled by Ikeuchi et al. [1].

Adult-onset leukoencephalopathy with axonal spheroid and pigmented glia (ALSP) is a common cause of adult-onset leukoencephalopathy [2]. In past decades, several reports have been published, and ALSP was proposed as a condition encompassing hereditary diffuse leukoencephalopathy with spheroids (HDLS) and familial pigmentary orthochromatic leukodystrophy (POLD) due to similar clinicopathological features [3,4] with the presence of colony stimulating factor-1 receptor (*CSF1R*) mutations. CSF1R is a tyrosine kinase receptor for colony stimulating factor-1 and interleukin-34, expressed in microglia and neurons. This receptor regulates microglial proliferation and survival and neuronal progenitor cells’ survival, proliferation, and differentiation. Over 120 ALSP cases have been reported to date based on *CSF1R* mutation [5]. In the Republic of Korea, several cases of ALSP with *CSF1R* mutation have been reported [6,7].

Patients with ALSP commonly present in their fourth or fifth decade, with a mean age of 43 (18−78) years [5]. Women have a younger age of onset (40 years) than men (47 years). Although the median life expectancy is 6.8 years, it varies considerably, with a survival period of over 20 years after the onset [5]. ALSP is characterized by cognitive decline, gradual behavioral changes, impaired movement, dominantly frontal subcortical white matter changes, and thinning of the corpus callosum. Further, an extensive loss of myelin and axons, abundant destructive axonal swelling, and pigmented macrophages characterize the neuropathological features of ALSP.

The clinical diagnostic criteria for ALSP suggested by Konno et al. [8] include 5 core features: (1) age at onset ≤ 60 years; (2) more than two of the following clinical signs and symptoms: cognitive impairment or psychiatric symptoms, pyramidal signs, parkinsonism, and epilepsy; (3) autosomal dominant inheritance or sporadic occurrence; (4) bilateral cerebral white matter lesions and thinning of the corpus callosum on brain computed tomography (CT)/magnetic resonance imaging (MRI); and (5) exclusion of other causes of leukoencephalopathy, including vascular dementia, multiple sclerosis, or leukodystrophy. *CSF1R* mutation is essential for a definitive diagnosis [8]. Moreover, Onayagi et al. proposed staging for ALSP disease progression through a review of autopsy cases [9].

Despite these efforts, making an accurate ALSP diagnosis is difficult due to its very low prevalence and various clinical features that require differentiation from those of other diseases. Therefore, when the clinician cannot confirm an ALSP diagnosis, the patient may undergo biopsy for a definitive diagnosis without genetic sequencing. In such cases, the pathologist may encounter these lesions on a biopsy or frozen specimen, as in our case 3, and should recognize that the numbers of axonal spheroids, pigmented cells, and white matter degeneration in a stereotactic brain biopsy specimen could vary depending on the biopsy site and stage of ALSP.

Herein, we describe the clinicopathological and genetic findings of three patients with ALSP.

## 2. Case Presentations

### 2.1. Case 1

The first patient was a 37-year-old woman who developed slowly progressive cognitive dysfunction and abnormal behaviors, such as apathy, aggression, and emotional disorders. She had difficulty performing simple housework. She developed stuttering, dysarthria, and an unstable gait. Her medical history included a left thyroidectomy for thyroid carcinoma at age 36 years. Two years after symptom onset, her Mini-Mental Status Examination (MMSE) score was 21 of 30. Neurological examination revealed dysarthria, bradykinesia, bilateral leg weakness, ataxia in the left upper extremity, hyperactive deep tendon reflexes in the left extremities, and bilateral extensor plantar responses. Over the following year, she developed paraparesis and became wheelchair-bound. Brain fluid-attenuated inversion recovery (FLAIR) MRI demonstrated high-signal intensity in the bilateral periventricular white matter (Figure 1A). Diffuse cortical and corpus callosal atrophy was noted. Her cerebrospinal fluid (CSF) and blood tests, including liver and thyroid function tests, complete blood count (CBC), and serum electrolyte levels, were within normal limits. Tests for lupus anticoagulant, rheumatoid factor, fluorescent antinuclear antibody (FANA), anti-double-stranded DNA antibody (Ab), anti-cardiolipin Ab, anti-Sjögren’s-syndrome-related antigen (SS) A, anti-SSB, serum antineutrophil cytoplasmic antibody (ANCA), anti-HIV Ab, venereal disease research laboratory test (VDRL), ceruloplasmin, lead, copper, lactic acid, and very long-chain fatty acids were negative. Ultrastructural examination of a skin biopsy specimen did not reveal evidence of cerebral autosomal dominant arteriopathy with subcortical infarcts and leukoencephalopathy (CADASIL). She was treated with steroids under the clinical suspicion of multiple sclerosis. However, her symptoms did not improve, and she died of septic shock at 41 years of age. A brain autopsy was performed to evaluate her brain disease since her elder brother began to show similar clinical manifestations. Her mother and two uncles had a history of dementia. This case has been previously reported [6].

#### Postmortem Findings

The brain weighed 1100 g, which was less than normal (1290 g, 41–50 years). Mild bilateral gyral atrophy of the frontal and parietal lobes was observed (Figure 1B). The temporal and occipital lobes and cerebellum were relatively preserved. Coronal and horizontal sections were obtained for the left and right hemispheres, respectively. The corpus callosum showed severe atrophy, with relative splenial sparing. The periventricular and deep white matter of the frontal and parietal lobes were severely degenerated and soft with discoloration (Figure 1C,D); however, the subcortical U-fibers were spared (Figure 1C–E). These gross findings were consistent with what the brain MRI revealed in aspect of frontotemporoparietal cortex and corpus callosum atrophy. Asymmetrical atrophy of the cerebral peduncle and pyramid was observed (Figure 1F–H). The basal ganglia, hippocampus, and cerebellum (Figure 1I) were relatively preserved.

Histologically, the white matter of the frontal and parietal lobes revealed scattered axonal spheroids and pigmented macrophages on hematoxylin-eosin (HE) stain (Figure 2A), and extensive loss of myelinated axons with sparing of subcortical U-fibers was confirmed by Luxol-fast blue (LFB) and Bielschovsky’s silver staining (BS) (Figure 2B). The loss of myelinated axons or pigmented macrophages affected the corpus callosum, internal capsule, and unilateral corticospinal tract (Figure 2C). Pigmented macrophages were positive for periodic acid-Schiff (PAS) and PAS with diastase, Sudan black B, Fontana-Masson, and LFB. Ballooning of degenerated cortical neurons was rarely observed in the frontal and parietal lobes. Immunohistochemically, the axonal spheroids and pigmented macrophages were positive for phosphorylated neurofilament (NF) (Figure 2D) and CD68 (Figure 2E). A few ballooned cortical neurons were also stained for phosphorylated NF (Figure 2F). Ultrastructurally, the pigmented granules in macrophages exhibited lipofuscin ceroid (Figure 2G). Postmortem pathological diagnosis was POLD in 2007, and later in 2015, a heterozygous c.2345G > A (p.R782H) pathogenic variant was identified in exon 18 of the *CSF1R* gene by direct sequencing of exons 12 to 20 using genomic DNA isolated from postmortem frozen peripheral blood leukocytes.

### 2.2. Case 2

A 44-year-old man presented with progressive cognitive impairment and hesitant speech. The patient was the elder brother of the patient in case 1. At the first evaluation, approximately 2 years after symptom onset, he scored 24/30 on the MMSE. Laboratory tests, including CBC, electrolytes, chemistry, thyroid and liver function tests, lupus anticoagulant, vitamin B12, folate, homocysteine, anti-SSA, anti-SSB, ANCA, FANA, lead, copper, lactic acid, CSF analyses, and Notch Receptor 3 gene, were normal or negative. Brain MRI showed high-signal intensities in the bilateral cerebral white matter on the T2 and FLAIR images (Figure 3A), which indicated the possibility of ischemic change.

However, the T2 signaling pattern was slightly perpendicular to the ventricle; therefore, demyelinating disease was included in the differential diagnosis. Glucose hypometabolism in the left parietal cortex, thalamus, and right cerebellum was revealed by ^18^F-fluoro-2-deoxy-D-glucose positron emission tomography. Speech difficulty became severe, and the patient showed depression, abulia, and bradykinesia. Two years after the first symptoms appeared, the patient could no longer perform his job. Brain CT was taken six years after the first brain MRI underwent. The brain CT revealed a more increased extent of hypodense lesion, which is compatible with high-signal intensities in the FLAIR sequence of MRI, in the white matter around the anterior horn of the bilateral lateral ventricle, and more progression of brain atrophy in the frontotemporoparietal lobes, with hydrocephalus ex vacuo. (Figure 3B). Over time, the symptoms worsened, and he showed magnetic gait and dysphagia. Five years before his death, he was paralyzed. He developed a seizure disorder, which was controlled with medicine, 3 years before his death at 54 years of age.

In exon 18 of the *CSF1R* gene, there was a heterozygous pathogenic variant (c.2345G > A, p.R782H), which was detected by direct sequencing of *CSF1R* using peripheral blood, as observed in the younger sister (case 1).

#### Postmortem Findings

The brain weighed 1140 g, which was less than normal (1410 g, 51–55 years). Grossly, moderate gyral atrophy of the bilateral frontal lobes, mild gyral atrophy of both parietal lobes, marked atrophy of the corpus callosum, and hydrocephalus ex vacuo were observed (Figure 3C,D). The white matter was severely degenerated in the frontal, parietal, and temporal lobes and a focal area of the occipital lobe (Figure 3E,F). Subcortical U-fibers were spared. These findings and neuroimages were remarkably consistent. The uncinate process, amygdala, and hippocampus were relatively preserved. The thalamus was atrophic, but there were no remarkable changes in the basal ganglia and cerebellum. Histologically, diffuse white matter degeneration involving the frontal, parietal, temporal, and occipital lobes revealed loss of axons and myelin, as confirmed by BS (Figure 3G) and LFB staining, respectively. Axonal spheroids and pigmented macrophages were sparsely observed in affected white matter lesions (Figure 3H).

### 2.3. Case 3

A 39-year-old woman had struggled with dysarthria and dysphagia for 2 years. Brain MRI (FLAIR sequence) revealed a white matter lesion with high-signal intensities involving the bilateral corona radiata and corticospinal tracts (Figure 4A,B). Brain atrophy and hydrocephalus were absent. Under the clinical suspicion of multiple sclerosis, she was treated with steroids; however, her symptoms were not relieved, and additional symptoms, such as bradykinesia, spasticity, and gait disturbance, appeared. ANCA and anti-aquaporin 4 Ab were negative. Nerve conduction, CSF, and blood examination results were within normal limits. A stereotactic brain biopsy was performed under the suspicion of a demyelinating disorder. The biopsy samples, consisting of cores of white matter measuring 0.4 × 0.3 cm, revealed pigmented macrophages and axonal spheroids (Figure 4D) and loss of myelinated axons (Figure 4F). A pathological diagnosis was ALSP. *Alanyl-transfer RNA synthetase 2* (*AARS2*) and *CSF1R* gene tests were performed to confirm ALSP. A heterozygous variant (c.1765G > A, p.G589R) was detected in exon 13 of the *CSF1R* gene. Intraventricular hemorrhage occurred because of a slip-down accident, and the right contraction became severe, making it difficult to point accurately. We continued follow-up in the outpatient clinic, and she had difficulty maintaining a sitting position and walking due to impaired balance. Her small-muscle stiffness worsened; consequently, she could not use the mobile phone and could only use a remote control, although she communicated correctly.

## 3. Discussion

We have described three cases (Table 1) of pathologically and genetically proven ALSP. The first and second cases were siblings diagnosed by postmortem examination, and the third case was diagnosed by pathologic examination of a stereotactic brain biopsy specimen.

All the patients presented with dysarthria, cognitive impairment, and bradykinesia. Histologically, all cases revealed cerebral white matter degeneration with diffuse myelin loss, axonal destruction, and a variable presentation of axonal spheroids and pigmented macrophages. The first case was previously described in a report of 4 Korean cases in 2015 [6]. The patient was initially diagnosed with POLD by postmortem brain examination in 2007 when her elder brother showed similar clinical presentations. The neuropathological findings were compatible with those of POLD, which is defined as myelin and axonal loss with pigmented macrophages. ALSP has been previously described as two separate entities: HDLS and POLD. Because there was considerable overlap in the morphologic findings of published cases of HDLS and POLD [3,4], Marrioti et al. proposed a single entity of ALSP for POLD and HDLS in 2009 [3]. Mutations in the tyrosine kinase domain (TKD) of the *CSF1R* gene were discovered in 2012 using whole-exome sequencing in 14 families affected by HDLS. Both POLD and HDLS are associated with *CSF1R* gene mutations. Genetic analyses of cases 1 and 2 for ALSP were available in 2016 and showed the same *CSF1R* pathogenic variant (c.2345G > A, p.R782H). This variant has been described in American POLD cases and Japanese families with HDLS [10,11].

ALSP diagnosis remains challenging because of the various clinical presentations that can mimic frontotemporal lobar degeneration, atypical parkinsonism, or primary progressive multiple sclerosis [4,12,13]. Thus, such a variety of differential diagnoses suggests that ALSP may be underdiagnosed. Some reports suggest that ALSP and *AARS2* mutation-related leukodystrophy (AARS2-L) are histopathologically similar [14]; however, the histopathology of AARS2-L is not fully understood because of the lack of autopsy-proven AARS2-L. AARS2-L usually presents with childhood- to adulthood-onset neurological deterioration, presenting with findings such as ataxia, spasticity, cognitive decline, and frontal lobe dysfunction, which are common findings in ALSP [15]. Previous reports have identified premature ovarian failure in female patients, periventricular white matter rarefaction with suppression of the FLAIR signal, and absence of periventricular calcification, which are important features of AARS2-L distinct from those of ALSP [14,16]. A previous study reported that calcifications are present in the white matter in ALSP [5]. However, the absence of premature ovarian insufficiency poses difficulty in distinguishing these two diseases. A recent autopsy report described a case of ALSP with premature ovarian failure [7].

Polycystic lipomembranous osteodysplasia with sclerosing leukoencephalopathy (Nasu–Hakola) disease is also characterized by pigmented microglia, leukoencephalopathy, and axonal spheroids. Nasu–Hakola disease is an autosomal recessive disorder and shows a relatively well-preserved internal capsule and pontine base, distinct from ALSP [9]. However, ALSP may be easily distinguished from Nasu–Hakola disease, which is characterized by common clinical findings, such as pain and swelling of wrists and ankles, and cystic bone lesions [17].

Lower motor neuron involvement is uncommon in inherited leukoencephalopathies, but it has been reported in association with Krabbe’s disease, Alexander disease, LMNB1-related autosomal dominant leukodystrophy, and adult-onset polyglucosan body disease. Even if specific neuropathological features are not observed when both upper and lower motor neuron symptoms are present, it is worth considering ALSP as a differential diagnosis [18,19].

Many other diseases can be discriminated, but clinicians can obtain assistance before biopsy through genetic testing. The classification of adult-onset leukoencephalopathies with genetic alterations was compiled by Ikeuchi et al. [1]. Although the predominant histology of these subtypes has been described, it is challenging to definitively diagnose without genetic alterations through DNA sequencing.

The early detection of a known *CSF1R* mutation may obviate the need for brain biopsy and its associated risks. Early genetic diagnosis in patients with ALSP will help clinicians determine prognosis, genetic risk in other family members, and the need for reproductive counseling within families. In addition, when a diagnostic brain biopsy has been performed, the biopsy sample should be carefully examined for axonal spheroids or pigmented glia, and consideration should be given to *CSF1R* gene sequencing.

Over 100 genetic alterations in *CSF1R* have been reported, including missense, nonsense, insertion/deletion, frameshift, and splice mutations. Most mutations are in the TKD of the CSF1R protein, encoded by exons 12–21 of the gene. Mutation hotspots were observed in exons 18−20 [20]. Mutations were more frequent in the distal TKD region encoded by exons 17−21 than in the proximal TKD region encoded by exons 12−15 [20]. As the number of cases increases, new locations are discovered; hence, it is necessary to check other locations through whole-genome sequencing.

The *CSF1R* variants detected in our cases were p.R782H (cases 1 and 2) and p.G589R (case 3). The p.G589R variant was described in a recent study on mutational analysis of *CSF1R* in 149 unrelated Taiwanese patients with adult-onset leukoencephalopathy [21]. In that study, five *CSF1R* variants (p.T79M, p.K586*, p.G589R, p.R777Q, and p.R782C) were identified. P.G589R and p.R777Q variants had been reported in patients with ALSP [2,5]. P.R782H and p.G589R variants are located within the TKD region of *CSF1R.* According to the American College of Medical Genetics and Genomics and the Association for Molecular Pathology (ACMG/AMP) guidelines [22], p.R782H and p.G589R variants are classified as pathogenic and likely pathogenic, respectively. After extensive review, there has been no clear correlation between genetic alterations and histological features of ALSP.

Onayagi et al. [9] proposed four stages of lesion severity in ALSP based on the degree of axonal loss: (1) stage I, myelinated axonal loss in scattered patchy areas of the centrum semiovale without cerebral atrophy; (2) stage II, myelinated axonal loss in large patchy areas of the cerebral white matter with slight cerebral atrophy, but preservation of cerebral subcortical U-fibers and the brainstem and cerebellum, and slight dilatation of the lateral and third ventricles; (3) stage III, extensive degeneration of cerebral white matter, including the corpus callosum, internal capsule, and some parts of the U-fibers; atrophy of the thalamus; and moderate dilatation of the lateral and third ventricles, with preservation of the brainstem/cerebellum; and (4) stage IV, devastated cerebral white matter with marked dilatation of ventricles and white matter degeneration of the brainstem/cerebellum. Cases 1 and 2 were classified as stage III ALSP. In both cases, the white matter degeneration and axonal loss were more severe and extensive through the cerebrum. However, axonal spheroids and pigmented macrophages were sparse in case 2, compared to case 1, who had many macrophages and axonal spheroids. This histological difference between cases 1 and 2 may reflect the duration of the disease (5 vs. 10 years). The age at symptom onset was lower (37 years) in case 1 than in case 2 (44 years), although both patients had the same genetic alterations.

Because the staging system is based on autopsy, there is a limit to its application based on tissue biopsy; however, it may be partially assessed by combining brain images. In case 3, the lesion stage seemed to be I or II based on neuroimaging, which showed T2 high-signal intensity in the corona radiata and bilateral corticospinal tract and mild dilatation of the lateral and third ventricles at the time of diagnosis.

We have presented three cases of pathologically and genetically confirmed ALSP and described their clinicopathologic and genetic findings in detail. Because all three patients were clinically suspected of having vascular dementia or a demyelinating disorder, we examined the brain autopsy and stereotactic biopsy results and provided a pathologic diagnosis of ALSP, followed by genetic confirmation. Therefore, ALSP should be included as a differential diagnosis in stereotactic biopsies or postmortem examinations of patients with adult-onset leukoencephalopathy.

## Figures and Tables

**Figure 1 diagnostics-13-01018-f001:**
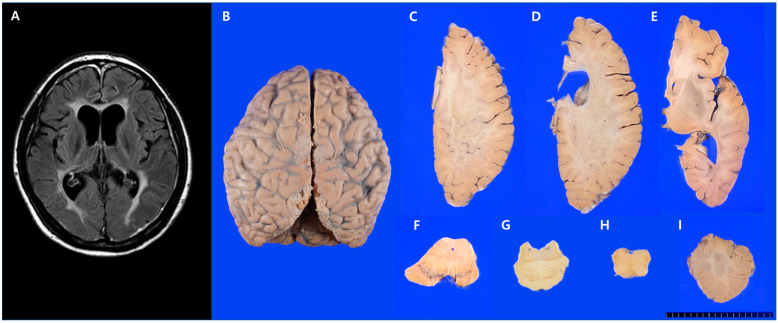
Brain magnetic resonance imaging (MRI) and gross findings of the brain in case 1. (**A**) Brain MRI (FLAIR sequence) revealed high-signal intensities in the bilateral periventricular white matter. (**B**) Mild gyral atrophy of the bilateral frontal and parietal lobes. (**C**–**E**) Horizontal sections of the right hemisphere revealed white matter degeneration of the frontal and parietal lobes, sparing of subcortical U-fibers, and severe corpus callosal atrophy. (**F**–**H**) There was symmetrical atrophy of the cerebral peduncle and pyramid and (**I**) a relatively preserved cerebellum.

**Figure 2 diagnostics-13-01018-f002:**
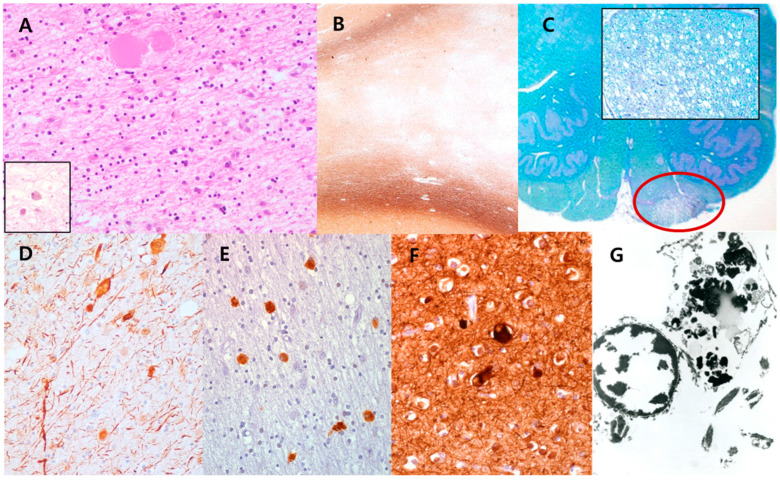
Histopathological findings of the brain in case 1. (**A**) The white matter lesions revealed scattered axonal spheroids and pigmented macrophages (inset) on hematoxylin-eosin stain (200×), and (**B**) extensive loss of myelinated axons with sparing of subcortical U-fibers on Bielschovsky’s silver staining (40×). (**C**) The unilateral corticospinal tract (red circle) was also affected by the loss of myelinated axons or pigmented macrophages (LFB, scan view; inset: LFB, 200×). Immunohistochemically, (**D**) axonal spheroids and pigmented macrophages were positive for phosphorylated neurofilaments (NF) (200×) and (**E**) CD68 (400×), respectively. (**F**) A few ballooned cortical neurons were stained with phosphorylated NF (400×). (**G**) Ultrastructurally, the pigmented granules in macrophages exhibited lipofuscin ceroids (7000×).

**Figure 3 diagnostics-13-01018-f003:**
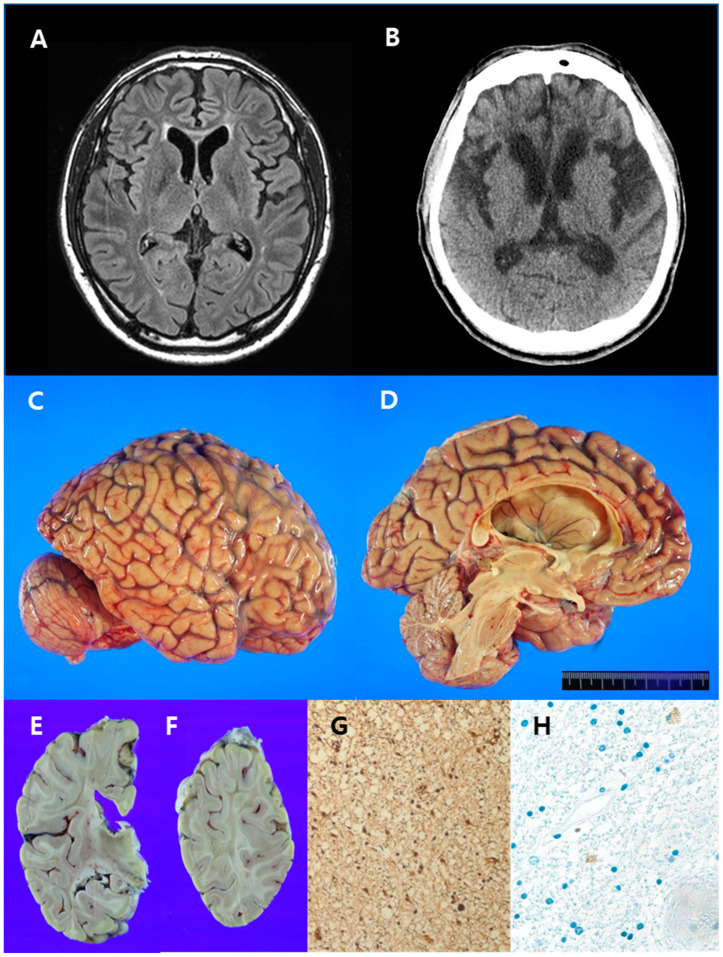
Radiological images, gross and histopathological findings of the brain in case 2. (**A**) Brain magnetic resonance imaging (FLAIR sequence) revealed high-signal intensities in the white matter around the anterior horns of the bilateral lateral ventricle. (**B**) Brain computed tomography was performed six years after the initial brain MRI and revealed hypodense lesions in the white matter around the anterior horns of the bilateral lateral ventricles and mild brain atrophy with hydrocephalus ex vacuo. (**C**,**D**) Findings included moderate gyral atrophy of the frontal lobe, mild gyral atrophy of the parietal lobe, marked atrophy of the corpus callosum, and hydrocephalus ex vacuo. (**E**,**F**) The white matter was severely degenerated in the frontal, parietal, and temporal lobes and in a focal area of the occipital lobe. Subcortical U-fibers were spared. (**G**) Severe axonal loss, but absent axonal spheroids, was noted (Bielschovsky’s silver stain, 200×). (**H**) Pigmented macrophages were sparsely observed (CD68, 200×).

**Figure 4 diagnostics-13-01018-f004:**
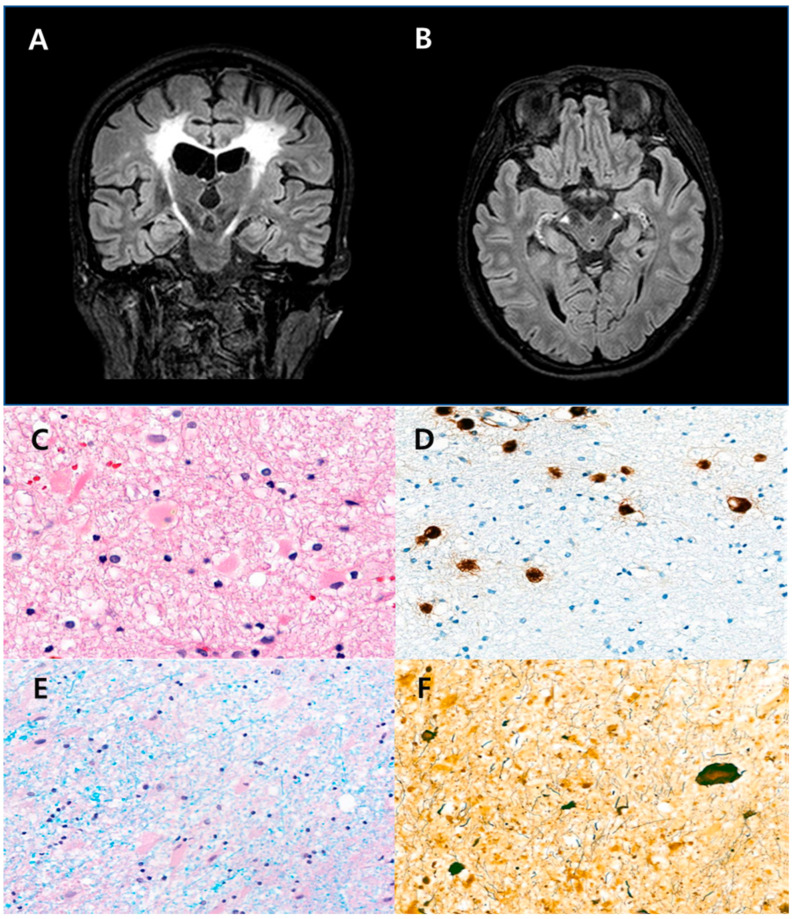
Brain magnetic resonance imaging (MRI) and histopathological findings of case 3. (**A**,**B**) Brain MRI (FLAIR sequence) revealed multifocal high-signal intensities in the bilateral corona radiata and corticospinal tracts. (**C**) There were frequent eosinophilic spheroids (hematoxylin-eosin stain, 200×) and (**D**) CD68-positive pigmented macrophages in the white matter (CD68, 100×). (**E**) Loss of myelinated (Luxol-fast blue stain, 100×) axons and (**F**) axonal spheroids were discovered in the white matter lesion (Bielschovsky’s silver stain, 100×).

**Table 1 diagnostics-13-01018-t001:** Summary of Cases.

		Case 1 *	Case 2 *	Case 3
		Autopsy	Autopsy	Stereotactic biopsy
Age at onset (years)		37	44	39
Sex		Female	Male	Female
Duration of disease		4 years	10 years	3 years
	Cognitive impairment	Present	Present	Present
	Psychiatric symptoms (e.g., depression, apathy, abulia, indifference, anxiety, irritability, distraction)	Present	Present	Present
	Pyramidal signs (hyperreflexia, spasticity, increased tone in extremities, pseudobulbar palsy)	Present	Present	Present
	Parkinsonism (resting tremor, rigidity, bradykinesia, postural instability)	Present	Present	Present
	Epilepsy	Absent	Present (at advanced stage)	Absent
Autosomal dominant (AD) inheritance or sporadic occurrence		AD	AD	Sporadic
	Bilateral cerebral white matter lesions	Present	Present	Present
	Thinning of the corpus callosum	Present	Present	Present
Stage (by Oyanagi)		II	III	N/A
*CSF1R* mutation (pathogenic variant)		c.2345G > A	c.2345G > A	c.1765G > A

* Cases 1 and 2 occurred in siblings.

## Data Availability

The data presented in this study are available in this article.

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
