# Peer review of "Adult-Onset Leukoencephalopathy with Axonal Spheroid and Pigmented Glia: Different Histological Spectrums Presented in Autopsy Cases of Siblings and a Surgical Case of Stereotactic Biopsy"

_diagnostics, 2023, doi:10.3390/diagnostics13061018_

Round 1

Reviewer 1 Report

The authors have presented a nice description of neuropathological findings of three patients with genetically-defined diagnosis of CSF1R-related leukoencephalopathy in the presentation of adult-onset leukoencephalopathy with axonal spheroids and pigmented glia. Some points should be discussed by the authors at this point: 

1. Have the authors evaluated in their study the existence of spheroid bodies in spinal or cortical motor neurons? I suggest including a brief discussion in case of positive findings in neuropathological studies, taking into account previous literature descriptions (e.g., Rev Neurol (Paris) 2020;176(3):219-221). 

2. Do the authors consider that stereotactic brain biopsy should be considered in the evaluation and diagnostic work-up of patients after genetic testing disclosing variants of uncertain significance (VUS) in the CSF1R gene? 

3. I suggest authors to change the use of the word "mutation" to pathogenic variant in several paragraphs of the manuscript. 

Author Response

Thank you for giving us the opportunity to submit a revised draft of the manuscript “Adult-onset leukoencephalopathy with axonal spheroid and pigmented glia: Different histological spectrums presented in autopsy cases of siblings and a surgical case of stereotactic biopsy” for publication in the Journal of Diagnostics. We appreciate the time and effort that the reiewers dedicated to providing feedback on our manuscript and are grateful for the insightful comments on and valuable improvements to our paper. Those changes are highlighted within the manuscript. Please see below, in red, for a point-by-point response to your comments and concerns.

Point 1: Have the authors evaluated in their study the existence of spheroid bodies in spinal or cortical motor neurons? I suggest including a brief discussion in case of positive findings in neuropathological studies, taking into account previous literature descriptions (e.g., Rev Neurol (Paris) 2020;176(3):219-221).

Response 1: Thank you for your comment. Unfortunately, the spinal cord was not taken for the pathologic examination because of the limited brain autopsy. But the cortical motor neurons revealed no specific changes except for ballooning of few cortical neurons. We added a following sentence in the discussion section of the revised manuscript. (Page 9 : line276-281)

Lower motor neuron involvement is uncommon in inherited leukoencephalopaties, but it has been reported in association with Krabbe's disease, Alexander' disease, LMNB1-related autosomal dominant leukodystrophy, and adult-onset polyglucosan body disease. Even if specific neuropathological features are not observed when both upper and lower motor neuron symptoms are present, it is considered worthy of considering ALSP as a differential diagnosis.

Point 2: Do the authors consider that stereotactic brain biopsy should be considered in the evaluation and diagnostic work-up of patients after genetic testing disclosing variants of uncertain significance (VUS) in the CSF1R gene?

Response 2: Thank you for your question. If pathologic findings of ALSP is found in the brain tissue of a patient who showed VUS in the CSF1R gene in a genetic test, further studies on whether the VUS is a disease-related gene is required. Therefore, it is thought that stereotactic biopsy on the brain tissue of such patients can help determine the meaning of the genetic mutation.

Point 3: I suggest authors to change the use of the word "mutation" to pathogenic variant in several paragraphs of the manuscript.

Response 3: Thank you for pointing this out. The word “mutation” has been changed to pathogenic variant in the revised manuscript.

Reviewer 2 Report

Congratulations to the authors for the results presented.

In case 1 and 2 the neuroradiological images are missing: can they be recovered?

Specify the correlation between neuroradiological and histological images.

Author Response

Thank you for giving us the opportunity to submit a revised draft of the manuscript “Adult-onset leukoencephalopathy with axonal spheroid and pigmented glia: Different histological spectrums presented in autopsy cases of siblings and a surgical case of stereotactic biopsy” for publication in the Journal of Diagnostics. We appreciate the time and effort that the reiewers dedicated to providing feedback on our manuscript and are grateful for the insightful comments on and valuable improvements to our paper. Those changes are highlighted within the manuscript. Please see below, in red, for a point-by-point response to your comments and concerns.

Point 1: In case 1 and 2 the neuroradiological images are missing: can they be recovered?

Specify the correlation between neuroradiological and histological images.

Response 1: Thank you for your comment. As you recommended, we have added neuroradiological images of cases 1 and 2 (Figure 1A, and Figure 3A and B) and a following sentences in the revised manuscript.

(Page 3: line120-121): These gross findings were consistent with what the brain MRI revealed in aspect of frontotemporoparietal cortex and corpus callosum atrophy.

(Page 6: line175-180): Brain CT was taken six years after the first brain MRI underwent. The brain CT revealed more increased extent of hypodense lesion, which is compatible with high signal intensities in FLAIR sequence of MRI, in the white matter around anterior horn of bilateral lateral ventricle and more progression of brain atrophy in frontotemporoparietal lobes with hy-drocephalus ex vacuo (Figure 3B).

(Page 6: line193): These findings and neuroimages were remarkably consistent.

We have added figure legends in the revised manuscript (figure 1A, and Figure 3A and B)

(Page 3 :Figure 1. Legend) Brain magnetic resonance imaging (MRI) and gross findings of the brain in case 1. (A) Brain MRI (FLAIR sequence) revealed high signal intensities in the bilateral periventricular white matter.

(Page 5: Figure 3. Legend) Radiological images, gross and histopathological findings of the brain in case 2. (A) Brain magnetic resonance imaging (FLAIR sequence) revealed high signal intensities in the white matter around anterior horn of bilateral lateral ventricle. (B) Brain computed tomography which performed six years after the initial brain MRI revealed hypodense lesions in the white matter around anterior horn of bilateral lateral ventricle and mild brain atrophy with hydrocephalus ex vacuo.

Reviewer 3 Report

The 3-case record is well written and comprehensively presented. The genetic testing supports the pathological diagnosis and properly the authors stress the relevance of the “early detection of a known pathogenic CSF1R mutation” with respect to biopsy. The paper adds relevant pieces of information, which might even modify the approach in a clinical setting.

Author Response

Response to Reviewer 3 Comments

Point 1: The 3-case record is well written and comprehensively presented. The genetic testing supports the pathological diagnosis and properly the authors stress the relevance of the “early detection of a known pathogenic CSF1R mutation” with respect to biopsy. The paper adds relevant pieces of information, which might even modify the approach in a clinical setting.

Respose 1: Thank you for your comment on our report “Adult-onset leukoencephalopathy with axonal spheroid and pigmented glia: Different histological spectrums presented in autopsy cases of siblings and a surgical case of stereotactic biopsy” for publication in the Journal of Diagnostics. We appreciate your efforts on review of our manuscript.

Round 2

Reviewer 2 Report

The corrections made to the manuscript and the integration with the neuroradiological images make the whole description of the scientific research more valid and understandable.